# ROBUST ANOMALY DETECTION AND BACKDOOR ATTACK DETECTION VIA DIFFERENTIAL PRIVACY

**Min Du, Ruoxi Jia, Dawn Song**
University of California, Berkeley
{min.du,ruoxijia,dawnsong}@berkeley.edu

## ABSTRACT

Outlier detection and novelty detection are two important topics for anomaly detection. Suppose the majority of a dataset are drawn from a certain distribution, outlier detection and novelty detection both aim to detect data samples that do not fit the distribution. *Outliers* refer to data samples within this dataset, while *novelties* refer to new samples. In the meantime, backdoor poisoning attacks for machine learning models are achieved through injecting poisoning samples into the training dataset, which could be regarded as "outliers" that are intentionally added by attackers. Differential privacy has been proposed to avoid leaking any individual's information, when aggregated analysis is performed on a given dataset. It is typically achieved by adding random noise, either directly to the input dataset, or to intermediate results of the aggregation mechanism. In this paper, we demonstrate that applying differential privacy can improve the utility of outlier detection and novelty detection, with an extension to detect poisoning samples in backdoor attacks. We first present a theoretical analysis on how differential privacy helps with the detection, and then conduct extensive experiments to validate the effectiveness of differential privacy in improving outlier detection, novelty detection, and backdoor attack detection.

## 1 INTRODUCTION

Given a dataset where most of the samples are from a certain distribution, outlier detection aims to detect the minorities in the dataset that are far from the distribution, while the goal of novelty detection is to detect newly observed data samples that do not fit the distribution. On the other hand, poisoning examples that are intentionally added by attackers to achieve backdoor attacks could be treated as one type of "outliers" in the training dataset. Using machine learning for outlier/novelty detection is typically to train a model that learns the distribution where the training data samples are drawn from, and the final trained model could give a high anomaly score for the outliers/novelties that deviate from the same distribution. In both cases, the machine learning model is not supposed to learn from the outliers in the training dataset. Unfortunately, deep learning models that contain millions of parameters tend to remember too much (Song et al. [2017]), and can easily overfit to rare training samples (Carlini et al. [2018]).

Protecting data privacy has been a major concern in many applications, because sensitive data are being collected and analyzed. Differential privacy has been proposed to "hide" certain input data from the output; that is, by looking at the statistical results calculated from input data, one cannot tell whether the input data contain a certain record or not. The way of applying differential privacy is to add random noise to the input data or the data analysis procedure, such that the output difference caused by the input difference can be hidden by the noise. A known fact is that differential privacy implies stability (Kasiviswanathan et al. [2011]). Particularly, a differentially private learning algorithm is stable in the sense that the model learned by the algorithm is insensitive to the removal or replacement of an arbitrary point in the training dataset (Bousquet & Elisseeff [2002]). When the training dataset contains a handful of outliers, the output model of a stable learning algorithm should be close to the one trained on the clean portion of the training set. Intuitively, compared with the model trained on contaminated dataset, the one trained on clean data could be better at distinguishing outliers from normal data. Therefore, differential privacy can potentially be leveraged to improve the

identification of outliers. This motivates us to apply differential privacy to anomaly detection and defense against backdoor attacks.

**Our contribution.** First, we present a theoretical explanation on why differential privacy can help to detect outliers from a training and testing dataset, as well as an analysis on the relationship between the number of outliers to detect and the amount of random noise to apply. Second, to demonstrate the effectiveness, we apply differential privacy to an autoencoder network trained on a constructed MNIST dataset with injected outliers, for both outlier detection and novelty detection, to show how much the utility could be improved with different amount of outliers and noise. Third, we apply differential privacy to a real-world task - Hadoop file system log anomaly detection. System log anomaly detection is an important topic in computer security. Our proposed method greatly improves upon the state-of-the-art system in this field. The results indicate that differential privacy is able to eliminate almost all the false negatives, and achieve significantly improved overall utility, compared with the current state-of-the-art work DeepLog (Du et al. [2017]). Finally, via a proof-of-concept experiment using MNIST dataset with injected poisoning samples, we show that the idea of outlier detection could be extended to backdoor attack detection, and that differential privacy is able to further improve the performance.

## 2 Preliminary

Given an input dataset and an aggregation mechanism, *differential privacy* (Dwork [2011]) aims to output the requested aggregation results, which are guaranteed not to reveal the participation of any individual data record. Formally, differential privacy is defined as below:

**Definition 1** (Differential privacy). *A randomized mechanism $\mathcal{M}$ applied to data domain $\mathbb{D}$ is said to be $(\epsilon, \delta)$-differentially private if for any adjacent datasets $d$, $d'$ in $\mathbb{D}$, and any subset of outputs $\mathcal{S} \subseteq Range(\mathcal{M})$, it holds that*

$$\Pr[\mathcal{M}(d) \in \mathcal{S}] \leq e^\epsilon \Pr[\mathcal{M}(d') \in \mathcal{S}] + \delta,$$

where $\epsilon$ stands for the privacy bound, and $\delta$ stands for the probability to break this bound.

The *adjacent* datasets $d$, $d'$ could be understood as two databases, where only one record differs, i.e., $\|d - d'\|_1 = 1$. Differential privacy guarantees that the difference between $d$ and $d'$ are not revealed through inspecting the outputs $\mathcal{M}(d)$ and $\mathcal{M}(d')$. Clearly, the closer $\epsilon$ is to 0, the more indistinguishable $\mathcal{M}(d)$ and $\mathcal{M}(d')$ are, and hence the stronger the privacy guarantee is.

A common approach to enforcing differential privacy for a function $f : \mathbb{D} \to \mathbb{R}$, is to add random Gaussian noise $\mathcal{N}(0, \sigma^2)$ to perturb the output in $\mathbb{R}$. The intuition is that, for given adjacent datasets $d$ and $d'$, one cannot tell whether the difference between $f(d)$ and $f(d')$ is incurred by the single record that differs in $d$ and $d'$, or by the random noise being applied. The magnitude of Gaussian noise needs to be tailored to the maximum difference between $f(d)$ and $f(d')$, which is formally defined as $\mathcal{L}_2$-sensitivity.

**Definition 2** ($\mathcal{L}_2$-sensitivity). *The $\mathcal{L}_2$-sensitivity for a function $f : \mathbb{D} \to \mathbb{R}$ is:*

$$\Delta = \max_{\substack{d, d' \in \mathbb{D} \\ \|d - d'\|_1 = 1}} \|f(d) - f(d')\|_2$$

The noise scale $\sigma$ to apply can be calculated as below (Dwork et al. [2014]).

**Theorem 1.** *To perturb a function with sensitivity $\Delta$ under $(\epsilon, \delta)$ - differential privacy, the minimum noise scale $\sigma$ of Gaussian mechanism is given by: $\sigma = \frac{\Delta}{\epsilon} \cdot \sqrt{2 \ln \frac{1.25}{\delta}}$, where $\epsilon \in (0, 1)$.*

**Deep learning with differential privacy (Abadi et al. [2016])** The procedure of deep learning model training is to minimize the output of a loss function, through numerous stochastic gradient descent (SGD) steps. Abadi et al. [2016] proposed a differentially private SGD algorithm that works as follows. At each SGD step, a fixed number of randomly selected training samples are used as a mini batch. For each mini batch training, the following two operations are performed to enforce differential privacy: 1) clip the norm of the gradient for each example, with a clipping bound $C$, to

limit the sensitivity of gradient; 2) sum the clipped per-example gradients and add Gaussian noise $\mathcal{N}(0, \sigma^2)$, before updating the model parameters. Abadi et al. [2016] further proposed a moment accounting mechanism which calculates the aggregate privacy bound when performing SGD for multiple steps. Differential privacy is immune to post-processing. Therefore, the output of the trained model for any queries enjoys the same privacy guarantee as the above SGD-based training process.

## 3 THE CONNECTION BETWEEN DIFFERENTIAL PRIVACY AND OUTLIER DETECTION

By definition, random noise added into model training for differential privacy hides the influence of a single record on the learned model. Intuitively, if applying differential privacy to the training process, the contribution of rare training examples will be hidden by random noise, resulting in a model that underfits the outliers. Such model will facilitate novelty and outlier detection because it will be less confident in predicting the atypical examples. In this section, we first present a theorem to precisely characterize the above intuition, and then analyze the relationship between the number of outliers in the training dataset and the amount of noise to apply.

**Notations**   Let $\mathcal{Z}$ be the sample space and $\mathcal{H}$ be the hypothesis space. The loss function $l : \mathcal{H} \times \mathcal{Z} \to \mathbb{R}$ measures how well the hypothesis $h \in \mathcal{H}$ explains a data instance $z \in \mathcal{Z}$. A learning algorithm $\mathcal{A} : \mathcal{Z}^n \to \mathcal{H}$ learns some hypothesis $\mathcal{A}(S)$ given a set $S$ of $n$ samples. For instance, in supervised learning problems, $\mathcal{Z} = \mathcal{X} \times \mathcal{Y}$, where $\mathcal{X}$ is the feature space and $\mathcal{Y}$ is the label space; $\mathcal{H}$ is a collection of models $h : \mathcal{X} \to \mathcal{Y}$; and $l(h, z)$ measures how well $h$ predicts the feature-label relationship $z = (x, y)$.

Let $S = \{z_1, \ldots, z_n\}$ be a set of independent samples drawn from an unknown distribution $\mathcal{D}$ on $\mathcal{Z}$. For a given distribution $\mathcal{D}$, an *oracle* hypothesis is the one that minimizes the expected loss:

$$h^* = \arg\min_h \mathbb{E}_{z \sim \mathcal{D}}[l(h, z)] \tag{1}$$

We define an outlier as a data instance that has significantly different loss from the population under the oracle hypothesis.

**Definition 3.** *We say $\tilde{z}$ is an outlier with regard to distribution $\mathcal{D}$ if*

$$l(h^*, \tilde{z}) - \mathbb{E}_{z \sim \mathcal{D}}[l(h^*, z)] \geq T \tag{2}$$

*where $T$ is a constant that depends only on $\mathcal{D}$.*

We will prove the usefulness of differential privacy to detect outliers for the classes of learning algorithms that produce hypotheses converging to the optimal hypothesis asymptotically pointwise. We define such learning algorithms to be *uniformly asymptotic empirical risk minimization* (UAERM).

**Definition 4.** *A (possibly randomized) learning algorithm $\mathcal{A}$ is UAERM with rate $\xi(n, \mathcal{A})$ if for any distribution $\mathcal{D}$ defined on the domain $\mathcal{Z}$, it holds that*

$$\forall z \qquad |\mathbb{E}_{\mathcal{S} \sim \mathcal{D}^n} \mathbb{E}_{h \sim \mathcal{A}(S)} l(h, z) - l(h^*, z)| \leq \xi(n, \mathcal{A}) \tag{3}$$

In the definition, we make it explicit that the rate $\xi(n, \mathcal{A})$ depends on the learning algorithm $\mathcal{A}$. For instance, if $\mathcal{A}$ is a differentially private learning algorithm, the rate will depend on the privacy parameters. In that case, with slight abuse of notation, we will denote the rate for a $(\epsilon, \delta)$-differentially private learning algorithm trained on $n$ data instances by $\xi(n, \epsilon, \delta)$.

Due to the nonconvexity of their loss functions, neural networks may not enjoy a useful, tight characterization of the learning rate. Thus, we will empirically verify that using noisy SGD to learn differentially private neural networks is UAERM. Moreover, as we will show in the experiment, $\xi(n, \epsilon, \delta)$ grows as privacy parameters $\epsilon$ and $\delta$ become smaller. Intuitively, this is because larger noise is required to ensure stronger privacy guarantees, which, on the other hand, slows down the convergence of the learning algorithm.

Without loss of generality, we assume that $0 \leq l(h, z) \leq 1$. The following theorem exhibits how the prediction performance of differentially private models on normal data will differ from outliers and connects the difference to the privacy parameters of the learning algorithm and the amount of outliers in the training data.

**Theorem 2.** *Suppose that a learning algorithm $\mathcal{A}$ is $(\epsilon, \delta)$-differentially private and UAERM with the rate $\xi(n, \epsilon, \delta)$. Let $S' = S \cup U$, where $S \sim \mathcal{D}^n$ and $U$ contains $c$ arbitrary outliers. Then*

$$\mathbb{E}_{h \sim \mathcal{A}(S')} l(h, \tilde{z}) - \mathbb{E}_{h \sim \mathcal{A}(S')} \mathbb{E}_{z \sim \mathcal{D}} l(h, z)$$

$$\geq T - 2 \left( \xi(n, \epsilon, \delta) + \sqrt{\frac{n(e^\epsilon - 1 + \delta)^2}{2} \log \frac{2}{\gamma}} + e^{c\epsilon} - 1 + c e^{c\epsilon} \delta \right) \tag{4}$$

*with probability at least $1 - \gamma$.*

The two terms in the left-hand side of (4) represent the model's prediction loss on outliers and normal test data drawn from $\mathcal{D}$, respectively. Due to the stochasticity of differentially private learning algorithms, the difference is characterized by the expectation taken over the randomness of the learned models. The theorem establishes a lower bound on the prediction performance difference between normal and outlier data. A larger difference indicates that identifying outliers will be easier.

The impact of privacy parameters on the lower bound manifests itself in two aspects. On one hand, stronger privacy guarantees (i.e., smaller $\epsilon$ and $\delta$) will require higher noise to be added into the training process, which increases the learning rate $\xi(n, \epsilon, \delta)$. On the other hand, increasing privacy level will improve the stability of the learning algorithm; the resulting models tend to ignore the outliers in the training set and become closer to the ones trained on completely clean data, thus making the outlier detection more effective. The second aspect is embodied by the fact that the terms except $\xi(n, \epsilon, \delta)$ in the parenthesis of the lower bound grow with $\epsilon$ and $\delta$. Therefore, the privacy parameters cannot be too large or too small in order to ensure optimal anomaly detection performance.

Moreover, the relationship between the right-hand side of (4) and $c$ indicates that the anomaly detection problem will be more difficult with more outliers in the training set. Dissecting the right-hand side of (4), we further observe that $c$ appears always in tandem with $\epsilon$. This implies that for larger number of outliers in the training set (i.e., $c$ is larger), we will need to tune down $\epsilon$ and $\delta$ to maintain the same novelty detection performance.

Last but not least, the definition of outliers in our paper is quite general—it does not make any assumptions about how the outliers are generated. Also, we do not make assumptions about whether these outliers are in training or test data. Therefore, our analysis can shed light on detecting various types of anomalies, including but not limited to outlier/novelty detection, backdoor detection, and noisy label detection. In the following experimental section, we will focus our evaluation on outlier/novelty detection and defense against backdoor attacks.

## 4 EXPERIMENTS

This section empirically evaluates the effectiveness of differential privacy in improving anomaly detection and backdoor attack detection. We call an outlier/novelty or a poisoning example as a *positive*, and other normal data samples as *negatives*. The metrics measured by each experiment include: false positive (FP), false negative (FN), Precision = TP / (TP+FP), Recall = TP / (TP+FN), Area under the receiver operating characteristic curve (AUROC) which is the area under the TPR-FPR curve, Area under the Precision - Recall curve (AUPR) which summarizes the Precision - Recall curve, as well as F-measure $= 2 \times$ Precision $\times$ Recall / ( Precision $+$ Recall ). The detailed explanations of all these measures could be found in (Wikipedia [2019b;a]; scikit-learn [2017a;b]).

### 4.1 OUTLIER DETECTION AND NOVELTY DETECTION WITH AUTOENCODERS

Autoencoder is a type of neural network that has been widely used for outlier detection and novelty detection. It contains an encoder network which reduces the dimension of the input data, and a decoder network which aims to reconstruct the input. Hence, the learning goal of autoencoders is to minimize the reconstruction error, which is consequently the loss function. Because the dimensionality reduction brings information loss, and the learning goal encourages to preserve the information that is common to most training samples, outliers that contain rare information could be identified by measuring model loss. In this section, with a varying amount of outliers and noise scale, we show how differential privacy would improve the utility of anomaly detection with autoencoders.

**Datasets.** We utilize MNIST dataset composed by handwritten digits 0-9, and notMNIST dataset (Kaggle [2017]), which contains letters A-J with different fonts. The original MNIST data contain $60,000$ training images, and $10,000$ test images, which we refer to as *MNIST-train* and *MNIST-test* respectively. The notMNIST data contain $10,000$ training images and $1,000$ test images, denoted as *notMNIST-train* and *notMNIST-test*. Based on these datasets, we intentionally construct training datasets with varying amount of injected outliers. Specifically, each training dataset is constructed with a particular outlier ratio $r_o$, such that the resulted dataset MNIST-OD-train($r_o$) contains $60,000$ images in total, where a percentage of $1 - r_o$ are from MNIST-train, and $r_o$ are from notMNIST-train. For each training dataset MNIST-OD-train($r_o$), a set of autoencoder models are trained with varying noise scale $\sigma$ applied for differential privacy. For an autoencoder model trained on dataset MNIST-OD-train($r_o$), outlier detection is thus to detect the $r_o \times 60,000$ outliers from MNIST-OD-train($r_o$). For novelty detection, we further construct a test dataset MNIST-ND-test, which is composed by the entire MNIST-test dataset and notMNIST-test dataset, a total of $11,000$ images. The goal of novelty detection is to identify the $1,000$ notMNIST-test images as novelties.

**Evaluation metrics.** To check whether a data sample is an outlier/novelty using autoencoders, the standard practice is to set a loss threshold based on training statistics, and any sample having a loss above this threshold is an outlier/novelty. To measure the performance under different thresholds, we use the AUPR score which is a threshold-independent metric. Compared with other metrics such as the AUROC score, the AUPR score is more informative when the positive/negative classes are highly unbalanced, e.g., for outlier detection where the ratio of outliers is extremely low. More experiment settings with both AUPR and AUROC metrics are in appendix which present similar observations.

**Set-up.** For autoencoders, the encoder network contains 3 convolutional layers with max pooling, while the decoder network contains 3 corresponding upsampling layers. For differential privacy, we use a clipping bound $C = 1$ and $\delta = 10^{-5}$, and vary the noise scale $\sigma$ as in (Abadi et al. [2016]). All models are trained with a learning rate of 0.15, a mini-batch size of 200 and for a total of 60 epochs.

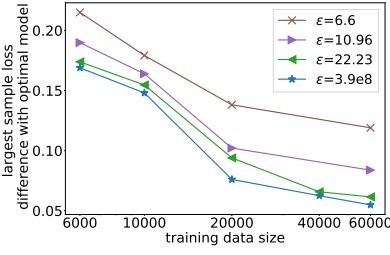

Figure 1: The largest test sample loss between a differentially private model trained on a random subset of training data and the oracle hypothesis.

| noise scale | \multicolumn{8}{c}{outlier percentage in training data $r_o$} |
|---|---|---|---|---|---|---|---|---|
| | \multicolumn{2}{c}{0.1%} | \multicolumn{2}{c}{0.5%} | \multicolumn{2}{c}{5%} | \multicolumn{2}{c}{10%} |
| $\sigma =$ | OD | ND | OD | ND | OD | ND | OD | ND |
| N/A | 99.92 | 99.77 | 92.12 | 98.81 | 84.33 | 88.18 | 72.16 | 68.14 |
| 0 | 99.89 | 99.83 | 98.3 | 99.69 | 83.86 | 87.91 | 77.8 | 74.74 |
| 0.01 | **100** | **99.97** | 94.92 | 99.23 | 90.79 | 93.34 | 85.41 | 84.07 |
| 0.1 | 100 | 99.85 | **98.44** | **99.66** | 92.23 | 94.21 | 85.56 | 83.98 |
| 1 | 100 | 99.78 | 98.28 | 99.67 | 94.92 | 96.87 | 81.87 | 80.12 |
| 5 | 99.87 | 99.49 | 98.51 | 99.52 | **96.78** | **98.04** | 95.25 | 95.41 |
| 10 | 90.24 | 97.77 | 91.88 | 98.2 | 96.6 | 98.2 | **97.07** | **97.46** |
| \multicolumn{9}{l}{Below $\sigma$ value is too big such that the model does not converge well.} |
| 50 | 65.94 | 92.13 | 70.34 | 90.8 | 86.58 | 91.59 | 88.49 | 90.27 |

Table 1: AUPR scores for outlier detection (OD) and novelty detection (ND). $\sigma = 0$ indicates applying clipping bound only.

**Validation of UAERM for Noisy SGD.** To begin with, we first conduct experiments to empirically validate our assumption in Theorem 2. While a rigorous verification of the assumption is intractable as it requires the knowledge of underlying data distribution and computing expected loss over randomness of both data distribution and differentially private algorithms, our experiments provide a sanity check of the assumption by replacing the expectation by the empirical average of a large number of data samples. For this set of experiments, we only utilize MNIST data for training, while the test dataset contains all available MNIST and notMNIST test samples. The oracle hypothesis is trained on all available training data, while each differentially private model is trained with varying privacy level $\epsilon$, and training data size. For a fixed training set and $\epsilon$, we perform training for multiple times to accommodate the randomness of differentially private training. Further, we train on multiple randomly selected training sets of the same size. We measure the loss of each resulting model on the test set, and calculate the average testing loss across different runs of differentially private training and different randomly selected training sets of the same size. We then compute the largest difference between the averaged test loss and the test loss associated with the oracle hypothesis. The results are shown in Figure 1. Each data point in the figure is an average of 9 differentially private models

trained on 3 randomly sampled subsets of the training data, and 3 random training processes for each sampled subset. As in Figure 1, the larger the training data size, and the larger $\epsilon$ is, the closer of the randomized model to the oracle hypothesis, validating our assumption in Theorem 2 that noisy SGD is UAERM.

**Detection results**   Table 1 shows the outlier detection (OD) results on dataset MNIST-OD-train, as well as the novelty detection (ND) results on dataset MNIST-ND-test. OD mimics the unsupervised anomaly detection case. ND mimics the case where the autoencoder model is supposed to be trained on normal data, to detect unforeseen anomalies, while the training dataset is noisy. The first row where $\sigma$ =N/A is for the baseline model without differential privacy applied. It performs well when $r_o = 0.1\%$, but drops significantly when $r_o$ reaches 0.5%. That's because for a mini-batch size of 200 that we use, an outlier ratio of $0.5\%$ in training data results in an average of one outlier in each mini-batch, which could be learned by the baseline model. Note that the clipping bound $C$=1 also restricts the contribution of outliers in SGD steps. We conduct an ablation study which only clips the per-example gradients with $C$ without adding any noise in each gradient descent step. The results are shown as $\sigma = 0$ in Table 1. As an intermediate step to bound the sensitivity in differential privacy, clipping itself is able to slightly improve the anomaly detection results in most cases. Still, we show that increasing the noise scale could further improve the utility. We highlight one of the best results in each column, and find that the trend follows our analysis in Theorem 2. Specifically, the more outliers in the training dataset, the larger noise scale is needed for the best improvement. As explained for (4), our theory shows that the privacy parameters cannot be too large or too small to ensure optimal anomaly detection performance, which coincides with the experimental results in Table 1. Although it could be challenging to select the desired noise level for training, we note that as shown in Table 1, applying differential privacy effectively improves the anomaly detection performance in most cases, except when $\sigma$ is too big to ruin the model parameters completely (e.g., $\sigma$=50) . Therefore, it is generally safe and almost always helpful to apply a small amount of differential privacy noise for anomaly detection. The noise scale could be increased further as long as the model converges. However, it should be noted that applying differential privacy makes the model training much slower than the baseline. In our experience utilizing NVIDIA Tesla V100 SXM2 GPU cards, the training time for each epoch could be up to 80 times longer. Finally, a training data portion as high as $10\%$ might not be "outliers", but could be part of the input pattern that should be learned by the model. We show in this case, a relatively large noise scale could effectively improve the anomaly detection results (e.g., $\sigma$=10), but it's up to the requirement of the application whether to apply this.

## 4.2 Hadoop file system log anomaly detection with LSTM

In this section, we use a real-world example for Hadoop file system log anomaly detection, to show how anomaly detection with differential privacy outperforms the current state-of-the-art results.

**Dataset**   The Hadoop file system (HDFS) log dataset (Wei Xu [2009]) is generated through running Hadoop map-reduce jobs for 48 hours on 203 Amazon EC2 nodes. This dataset contains over 11 million log entries, which could be further grouped into $575,059$ block sessions by the block identifier each log has. Each block is associated with a normal/abnormal label provided by domain experts. Over the past decade this log dataset has been extensively used for research in system log anomaly detection (Xu et al. [2009]; Lou et al. [2010]; Du et al. [2017]). The state-of-the-art results are achieved by DeepLog (Du et al. [2017]), which we use as the baseline model. As in DeepLog, our training dataset contains $4,855$ normal block sessions, while the test dataset includes $553,366$ normal sessions and $16,838$ abnormal sessions.

**Baseline model and metrics**   DeepLog utilizes LSTM neural networks to learn system log sequences. Note that system log messages are textual logs, e.g., *"Transaction A finished on server B."*. Before applying LSTM, a log parsing step first maps each log message into its corresponding log printing statement in the source code, e.g., *"print('Transaction %s finished on server %s."%(x,y))"*. Since there are only a constant number (e.g., $N$) of log printing statements in the source code, each one could be mapped to a discrete value from a fixed vocabulary set (e.g., having size $N$). With that, a block session of log messages could be parsed to a sequence of discrete values, e.g. *"22 5 5 5 11 9 11 9 11 9 26 26 26"*. Leveraging the fact that hidden execution paths written in source code restrict the possibilities of how one system log follows another, DeepLog trains an LSTM model

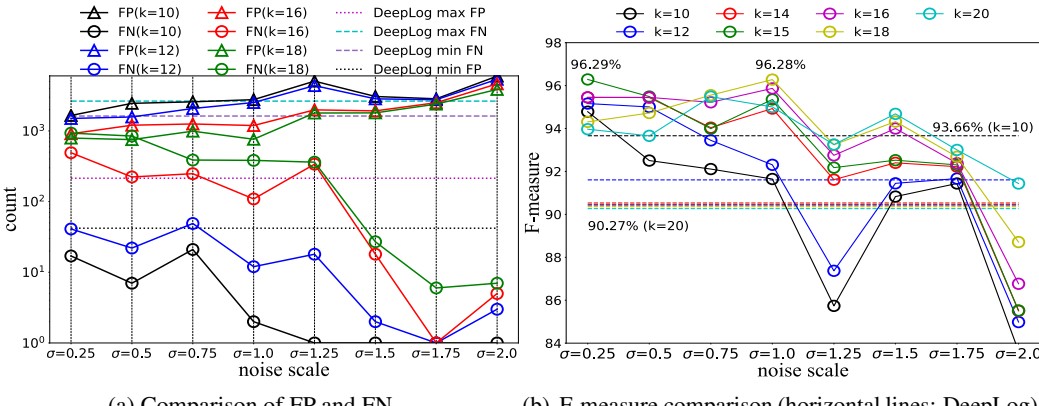

(a) Comparison of FP and FN .

(b) F-measure comparison (horizontal lines: DeepLog).

Figure 2: Improvements by differential privacy for DeepLog.

on normal discrete sequences, which learns to predict the next discrete value given its history. In detection, the LSTM model predicts a probability distribution on all possible values that may appear at a given time step. The real executed value is detected as abnormal if it's *unlikely* to happen based on LSTM prediction. The criteria presented in DeepLog is to first sort the predicted values based on the assigned probabilities, e.g., for a prediction "*{5: 0.2, 9: 0.08, 11: 0.01, 26: 0.7, ...}*", the order would be *26, 5, 9, 11, ...*. The given value to detect is checked against the sorted top $k$ predictions, and is detected as abnormal if it's not one of them. For anomaly detection metrics, we want to highlight that applying differential privacy significantly reduces false negatives, without introducing many false positives. Therefore, we'll focus on the comparison over the number of false positives and false negatives, while also presenting measurements that indicate the overall detection performance.

**Set up**   For the baseline model DeepLog, we train an LSTM model for 100 epochs, and use the final model as the anomaly detection model. The model related parameters are: 2 layers, 256 units per layer, 10 time steps, and a batch size of 256. We call the DeepLog model with differential privacy as *DeepLog+DP*. For differential privacy, we use a clipping bound $C = 1$, $\delta = 10^{-5}$, and vary the noise scale $\sigma$. All other model related settings for DeepLog+DP are the same as DeepLog.

**Results**   Figure 2a shows the comparison of FP and FN under different thresholds $k$, with the increase of noise scale $\sigma$. For clarity, we only show the following two cases for baseline model DeepLog: $k = 10$ which has the maximum FP and the minimum FN , and $k = 18$ which has the minimum FP and the maximum FN . Note that y axis is plotted as log scale. It is clear that applying DP noise significantly reduces FN in all cases, from over a thousand in DeepLog, to hundreds or even zero in DeepLog+DP. Also, the larger noise being added, the more FN are reduced. Although more FP could be introduced in some cases, we note that in system anomaly detection, the merit of fewer false negatives in fact worth the cost of more false positives. Reported false positives could be further checked by system admin, and then fed into the model for incremental learning. However, a false negative may never be found out, until a more disastrous event occurs due to the un-discovery of it.

The F-measure measurements are plotted in Figure 2b. For DeepLog model, F-measure ranges from 90.38% ($k = 20$) to 93.81% ($k = 10$). For DeepLog+DP, the best F-measure scores include 96.29% ($\sigma = 0.25$, $k = 15$) and 96.28% ($\sigma = 1$, $k = 18$), which show clear improvements over DeepLog model. Note that the best FN and FP measurements reported in DeepLog (Du et al. [2017]) are 619 and 833 respectively, while DeepLog+DP achieves FN =383, FP =762 at the F-measure of 96.28% ($\sigma = 1$, $k = 18$); and FN =123, FP =1040 at the F-measure of 96.29% ($\sigma = 0.25$, $k = 15$), showing its ability to significantly reduce false negatives without introducing many false positives. As shown in the figure, DeepLog performs better when $k$ is smaller, while DeepLog+DP benefits from larger $k$s. This scenario could also be explained by the addition of differential privacy noise. Since the trained model does not overfit to outliers, it assigns to anomalies much lower probabilities, so that anomalies are ranked much lower than that in the DeepLog model. Meanwhile, normal execution logs are also possibly predicted with lower probabilities because of the uncertainty brought by the noise. As a result, the ideal threshold $k$ for DeepLog+DP is higher than that of DeepLog. We also

note that a large noise scale could hurt the overall performance, as shown by the downward trend when $\sigma$ increases from 1.75 to 2.0.

## 4.3 BACKDOOR ATTACK DETECTION

Since poisoning examples for backdoor attacks are essentially "outlier" training samples injected by attackers, this section conducts proof-of-concept experiments to examine whether measuring model loss as for outliers works to detect poisoning examples, and whether differential privacy is able to further improve the performance. This detection scenario is particularly useful for backdoor attacks injected in the crowdsourcing scenario, where the model trainer gathers training data from untrusted individuals. In this case, the model trainer does not have control over the data quality but does have control over the model training process. Our proposal of adding DP noise is useful for detecting backdoor attacks and training more robust models in such a scenario.

**Dataset and set up**   MNIST dataset as described in Section 4.1 is used in this set of experiments. We refer the original 60,000 training images as *CLEAN-train* and the 10,000 test images as *CLEAN-test*. We construct the backdoor attacks as described in (Gu et al. [2017]), Section 3.1.2. Specifically, each poisoning example is generated by reversing 4 pixel values in the bottom right corner of a clean image having label $i$, and assigning backdoor label $(i+1)\%10$. From CLEAN-train, we randomly sample a poisoning ratio of $r_p$ images to be poisoning examples, resulting in a poisoned training dataset *POISONED-train($r_p$)*. To demonstrate the effectiveness of the poisoning attacks, we use the CLEAN-test dataset, as well as *POISONED-test* dataset which is constructed by poisoning *all* images in CLEAN-test. For image classification model, we use convolutional neural network (CNN) containing 2 convolutional layers with max pooling, and a softmax layer to output desired labels. The differentially private models are trained with the same configurations as in Section 4.1 unless otherwise noted.

**Metrics**   We first evaluate the effectiveness of the constructed backdoor attack. A successful backdoor attack should have high image classification accuracy on CLEAN-test, which we refer to as *benign accuracy*, as well as high accuracy on POISONED-test with poisoned labels, which indicates the *success rate*. To investigate whether measuring the classification model loss is able to detect poisoning examples, and whether differential privacy is able to improve the detection performance, we leverage metrics AUPR and AUROC as described at the beginning of Section 4.

| noise scale | detection (AUPR / AUROC) and attack (benign accuracy / success rate) performance | | | | | |
| | $r_p$ =0.5% | | $r_p$ =1% | | $r_p$ =5% | |
| $\sigma =$ | detection | attack | detection | attack | detection | attack |
| N/A | 73.01 / 99.26 | **98.93 / 47.85** | 27.02 / 95.23 | **98.95 / 97.12** | 14.85 / 78.88 | **99.11 / 98.1** |
| 0 | 91.22 / 99.92 | 97.66 / 0.23 | 92.11 / 99.88 | 97.84 / 0.35 | 95.33 / 99.79 | 97.46 / 0.3 |
| 0.005 | **92.64 / 99.9** | 97.57 / 0.17 | 94.04 / 99.93 | 97.46 / 0.28 | 94.76 / 99.79 | 97.75 / 0.3 |
| 0.01 | 92.24 / 99.92 | 97.51 / 0.25 | 94.03 / 99.92 | 97.4 / 0.34 | 93.4 / 99.74 | 97.55 / 0.31 |
| 0.05 | 90.76 / 99.9 | 97.42 / 0.24 | **95.11 / 99.94** | 97.8 / 0.37 | 95.09 / 99.83 | 97.72 / 0.3 |
| 0.1 | 92.16 / 99.93 | 97.55 / 0.25 | 94.85 / 99.93 | 97.7 / 0.28 | **95.33 / 99.82** | 97.34 / 0.39 |

Table 2: Backdoor attack and detection results with varying poisoning ratio $r_p$ (clipping bound $C = 1$).

**Results**   We first evaluate the backdoor attack effectiveness and the detection performance with varying poisoning ratio $r_p$, under different noise scale $\sigma$, and fixed clipping bound $C = 1$. The results are summarized in Table 2. $\sigma$ =N/A indicates classification models trained without differential privacy. Benign accuracy remains high on clean data. Backdoor success rate is only around half at a poisoning ratio of 0.5%, and shows successful (97.12% success rate) at a poisoning ratio of 1%. Detecting poisoning examples by measuring model loss shows some level of effectiveness when the poisoning ratio is low (e.g., 0.5%). Furthermore, applying differential privacy to the model training process is able to significantly improve the detection performance. Similar as in Table 1, the higher the poisoning ratio, the larger the noise level (smaller $\epsilon$) to achieve the best improvement. Another interesting observation is that, a differentially private model is naturally robust to backdoor attacks. As indicated in Table 2, differential privacy effectively limits the success of backdoor attacks, reducing the success rate below 0.5% in *all* cases. In comparison, the utility downgrade on benign accuracy is little.

To further evaluate the applicability of using the same CNN model for both anomaly detection and image classification, seeking to co-optimize the performance of the model for both tasks, we collect measurements with varying clipping bound $C$, and fixed noise scale $\sigma = 0.5$, as a complement to Table 2. The results are summarized in Table 3. Note that a small $C$ may hurt model performance as more model parameters are clipped. When $C$ is greater than most parameter values, the effect of increasing $C$ is similar to that of increasing $\sigma$ (Abadi et al. [2016]). From Table 3, we can observe that the best model for anomaly detection could have a similar set of parameters with the best model for image classification. However, in general, as shown in Table 2, classification accuracy and robustness are often two conflicting desiderata; model trainers can tune the privacy parameter in order to meet the task-specific requirements for accuracy and robustness.

| clipping bound | detection (AUPR / AUROC) and attack (benign accuracy / success rate) performance | | | | | |
| | $r_p = 0.5\%$ | | $r_p = 1\%$ | | $r_p = 5\%$ | |
| $C =$ | detection | attack | detection | attack | detection | attack |
|---|---|---|---|---|---|---|
| 0.5 | 87.46 / 99.87 | 96.29 / 0.31 | 90.78 / 99.85 | 96.47 / 0.26 | 95.62 / 99.79 | 96.73 / 0.34 |
| 0.8 | 89.03 / 99.85 | 97.13 / 0.33 | 92.39 / 99.89 | 97.11 / 0.32 | **95.63 / 99.79** | **97.4** / 0.28 |
| 1 | 90 / 99.9 | 97.28 / 0.22 | 93.46 / 99.92 | **97.47** / 0.25 | 95.37 / 99.79 | 97.34 / 0.3 |
| 2 | **90.85 / 99.81** | **97.48** / 0.24 | **93.49 / 99.91** | 97.21 / 0.26 | 93.26 / 99.75 | 97.39 / 0.46 |
| 3 | 90.17 / 99.93 | 97.29 / 0.3 | 88.05 / 99.84 | 97.18 / 0.33 | 89.51 / 99.59 | 97.35 / 0.48 |

Table 3: Backdoor attack and detection results with varying poisoning ratio $r_p$ (noise scale $\sigma = 0.5$).

## 5  RELATED WORK

To the best of our knowledge, this paper is the first one that proposes to improve outlier/novelty detection with differential privacy, and further extends it to backdoor attack detection. Note that this is not the first work that combines outlier detection and differential privacy together. Okada et al. [2015] aim to preserve input data privacy while detecting outliers. The two tasks are contradicting in this case as the identification of outliers (part of input data) implies certain privacy leakage, so Okada et al. [2015] try to find a balance. In contrast, we focus on improving anomaly detection performance with differential privacy, which is only applied to the model training stage, but no privacy protection is provided for the input data in detection stage when the outliers are actually being detected.

Outlier detection and novelty detection are closely related to each other and often addressed together (Hodge & Austin [2004]; scikit-learn [2017c]; Pedregosa et al. [2011]). Outlier detection is the process of identifying rare items in a dataset that significantly differ from the majority (Aggarwal & Yu [2001]), while novelty detection is to detect new observations that lie in the low density area of the existing dataset (Markou & Singh [2003a;b]). Previous work mostly achieves outlier detection using unsupervised learning methods (Zimek et al. [2012; 2014]; Campos et al. [2016]), while novelty detection typically assumes a normal dataset is available for training, and is realized by semi-supervised learning (Blanchard et al. [2010]; De Morsier et al. [2013]). In both cases, it involves summarizing a distribution that the majority of training data are drawn from. Traditional methods such as clustering (Duan et al. [2009]) and principal component analysis (PCA) (Xu et al. [2010]; Hoffmann [2007]) have been frequently used. In this paper, we leverage deep learning based detection methods including autoencoders (Gottschlich et al. [2017]) and LSTM (Du et al. [2017]) as the baselines, and further extend the idea of measuring model loss to backdoor attack detection.

Proposed by Dwork [2008], differential privacy has been a powerful tool to protect input data privacy. Kasiviswanathan et al. [2011] show that differential privacy implies stability on the output statistical results. Further, Dwork et al. [2015] point out that the empirical average of the output of a differentially private algorithm on a random dataset is close to the true expectation with high probability. Differential privacy has been utilized to train machine learning models that are robust to adversarial examples (Phan et al. [2019]; Lecuyer et al. [2018]), and to bound the success of inference attacks (Yeom et al. [2018]). In this paper, we utilize the property of differential privacy to improve anomaly detection and privacy is ensured via the technique proposed in Abadi et al. [2016].

Lastly, we note that a recent paper by Bagdasaryan & Shmatikov [2019] showed that accuracy of differentially private models drops much more for the underrepresented classes and subgroups. Intrinsically, our paper exploited the same phenomenon to improve anomaly detection. Bagdasaryan & Shmatikov [2019] studied the phenomenon empirically, while our work provides a theoretical analysis, which, for the first time, precisely characterizes the dependence of the performance gap between

the majority and the underrepresented group on the privacy parameters. Moreover, Bagdasaryan & Shmatikov [2019] mainly considered the implication of differential privacy to the fairness of machine learning models; by contrast, our paper focuses on anomaly detection and backdoor attacks and exhibits strong empirical evidence for the efficacy of differential privacy in these two application domains.

## 6 CONCLUSION

In this paper, inspired by the fact that differential privacy implies stability, we apply DP noise to improve the performance of outlier detection and novelty detection, with an extension to backdoor attack detection. We first provide the theoretical basis for the efficacy of differential privacy for identifying anomalies, connecting the hardness of the identification problem to privacy parameters. Our theoretical results are useful to explain various experimental findings, including how the anomaly detection performance varies with privacy parameters and the number of outliers in the training set. We perform extensive experiments to demonstrate the effectiveness of differential privacy for anomaly detection. To fully evaluate the effectiveness of DP in anomaly detection with different amount of outliers and noisee, we first construct a contaminated dataset based on MNIST and train autoencoder anomaly detection models with varying noise scale applied. We then evaluate the performance using a real-world task, Hadoop file system log anomaly detection, by applying DP noise to DeepLog, the current state-of-the-art detection model. The evaluation results show that DP noise is effective towards reducing the number of false negatives, and further improving the overall utility. Finally, we generalize the idea of measuring model loss for outlier detection to backdoor attack detection and further improve the performance via differential privacy.

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

## A  APPENDIX

### A.1  PROOF OF THEOREM 2

The following result will be used for reasoning about the performance gap of the differentially private learned models between regular data points and outliers.

**Theorem 3** (McDiarmid, 1989). *Let $S$ be a set of $n$ points and $S^i$ be the set with the $i$th element in $S$ replaced by $z_i'$, let $F : \mathcal{Z}^n \to \mathbb{R}$ be any measurable function for which there exits constants $c_i$ ($i = 1, \ldots, n$) such that*

$$\sup_{S \in \mathcal{Z}^n, z_i' \in \mathcal{Z}} |F(S) - F(S^i)| \le c_i \tag{5}$$

*then*

$$P_S[F(S) - \mathbb{E}_S[F(S)] \ge \epsilon] \le e^{-2\epsilon^2 / \sum_{i=1}^n c_i^2} \tag{6}$$

Moreover, we will need the following lemma on group differential privacy.

**Lemma 1.** *If $\mathcal{A}$ is $(\epsilon, \delta)$-differentially private with respect to one change in the database, then $\mathcal{A}$ is $(c\epsilon, ce^{c\epsilon}\delta)$-differentially private with respect to $c$ changes in the database.*

Now, we are ready to present the proof of Theorem 2.

**Theorem 2.** *Suppose that a learning algorithm $\mathcal{A}$ is $(\epsilon, \delta)$-differentially private and UAERM with the rate $\xi(n, \epsilon, \delta)$. Let $S' = S \cup U$, where $S \sim \mathcal{D}^n$ and $U$ contains $c$ arbitrary outliers. Then*

$$\mathbb{E}_{h \sim \mathcal{A}(S')} l(h, \tilde{z}) - \mathbb{E}_{h \sim \mathcal{A}(S')} \mathbb{E}_{z \sim \mathcal{D}} l(h, z)$$

$$\ge T - 2\left( \xi(n, \epsilon, \delta) + \sqrt{\frac{n(e^\epsilon - 1 + \delta)^2}{2} \log \frac{2}{\gamma}} + e^{c\epsilon} - 1 + ce^{c\epsilon}\delta \right) \tag{4}$$

*with probability at least $1 - \gamma$.*

*Proof.* Let the probability density/mass defined by $\mathcal{A}(S')$ and $\mathcal{A}(S)$ be $p(h)$ and $p'(h)$, respectively. Using Lemma 1, for any $z \in \mathcal{Z}$ we have

$$\mathbb{E}_{h \sim \mathcal{A}(S)} l(h, z) = \int_0^1 P_{h \sim \mathcal{A}(S)}[l(h, z) > t] dt \tag{7}$$

$$\le \int_0^1 (e^{c\epsilon} P_{h \sim \mathcal{A}(S')}[l(h, z) > t] + ce^{c\epsilon}\delta) dt \tag{8}$$

$$= e^{c\epsilon} \mathbb{E}_{h \sim \mathcal{A}(S')}[l(h, z)] + ce^{c\epsilon}\delta \tag{9}$$

It follows that

$$\mathbb{E}_{h \sim \mathcal{A}(S)} l(h, z) - \mathbb{E}_{h \sim \mathcal{A}(S')}[l(h, z)] \le (e^{c\epsilon} - 1)\mathbb{E}_{h \sim \mathcal{A}(S')}[l(h, z)] + ce^{c\epsilon}\delta \tag{10}$$

$$\le e^{c\epsilon} - 1 + ce^{c\epsilon}\delta \tag{11}$$

By symmetry, it also holds that

$$\mathbb{E}_{h \sim \mathcal{A}(S')} l(h, z) - \mathbb{E}_{h \sim \mathcal{A}(S)}[l(h, z)] \le (e^{c\epsilon} - 1)\mathbb{E}_{h \sim \mathcal{A}(S)}[l(h, z)] + ce^{c\epsilon}\delta \tag{12}$$

$$\le e^{c\epsilon} - 1 + ce^{c\epsilon}\delta \tag{13}$$

Thus, we have the following bound:

$$|\mathbb{E}_{h \sim \mathcal{A}(S)} l(h, z) - \mathbb{E}_{h \sim \mathcal{A}(S')}[l(h, z)]| \le e^{c\epsilon} - 1 + ce^{c\epsilon}\delta \tag{14}$$

Moreover, let $S^i$ be the set with the $i$th element in $S$ replaced by $z_i'$. Then, by the same argument above, we have

$$|\mathbb{E}_{h \sim \mathcal{A}(S)} l(h, z) - \mathbb{E}_{h \sim \mathcal{A}(S^i)}[l(h, z)]| \le e^\epsilon - 1 + \delta \tag{15}$$

for all $i = 1, \ldots, n$. Then, using Theorem 3 that relates the first order differences of random functions to their variance, we obtain

$$P_{S \sim \mathcal{D}}[|\mathbb{E}_{h \sim \mathcal{A}(S)} l(h, z) - \mathbb{E}_S \mathbb{E}_{h \sim \mathcal{A}(S)} l(h, z)| \geq \tau] \leq 2e^{-\frac{2\tau^2}{n(e^\epsilon - 1 + \delta)^2}} \tag{16}$$

Hence,

$$P_{S \sim \mathcal{D}}[|\mathbb{E}_{h \sim \mathcal{A}(S)} l(h, z) - \mathbb{E}_S \mathbb{E}_{h \sim \mathcal{A}(S)} l(h, z)| \geq \sqrt{\frac{n(e^\epsilon - 1 + \delta)^2}{2} \log \frac{2}{\gamma}}] \leq \gamma \tag{17}$$

Combining (14) with (17), we have

$$P_{S \sim \mathcal{D}}[|\mathbb{E}_{h \sim \mathcal{A}(S')} l(h, z) - \mathbb{E}_S \mathbb{E}_{h \sim \mathcal{A}(S)} l(h, z)| \leq \sqrt{\frac{n(e^\epsilon - 1 + \delta)^2}{2} \log \frac{2}{\gamma}} + e^{c\epsilon} - 1 + ce^{c\epsilon}\delta]$$
$$\geq 1 - \gamma \tag{18}$$

Since $\mathcal{A}$ is UAERM, the following inequality holds with probability at least $1 - \gamma$:

$$|\mathbb{E}_{h \sim \mathcal{A}(S')} l(h, z) - l(h^*, z)| \leq \xi(n, \epsilon, \delta) + \sqrt{\frac{n(e^\epsilon - 1 + \delta)^2}{2} \log \frac{2}{\gamma}} + e^{c\epsilon} - 1 + ce^{c\epsilon}\delta \tag{19}$$

for all $z \in \mathcal{Z}$.

For a given outlier $\tilde{z}$, it satisfies $l(h^*, \tilde{z}) - \mathbb{E}_{z \sim \mathcal{D}}[l(h^*, z)] \geq T$ by definition. Combining it with (19), we get

$$\mathbb{E}_{h \sim \mathcal{A}(S')} l(h, \tilde{z}) - \mathbb{E}_{z \sim \mathcal{D}} \mathbb{E}_{h \sim \mathcal{A}(S')} l(h, z)$$
$$\geq T - 2\left( \xi(n, \epsilon, \delta) + \sqrt{\frac{n(e^\epsilon - 1 + \delta)^2}{2} \log \frac{2}{\gamma}} + e^{c\epsilon} - 1 + ce^{c\epsilon}\delta \right) \tag{20}$$

$\square$

## A.2 ADDITIONAL EXPERIMENT RESULTS

We list additional experimental results in this section. With more parameters being tested and more metrics being collected, these extensive results further validate our observations presented in the main paper body: differential privacy could improve anomaly detection and backdoor attack detection; and the higher ratio of outliers in the training data, the more noise (smaller privacy bound $\epsilon$) is needed to achieve the best improvement.

### A.2.1 AUTOENCODER ANOMALY DETECTION

Besides the AUPR scores presented in Table 1 for outlier detection and novelty detection using autoencoders, we present additional results with more parameters being tested in Table 4. The observations are similar. As an intermediate step in differential privacy to bound the sensitivity, clipping itself without adding any noise is able to improve the performance of outlier detection and novelty detection. Adding various amounts of random Gaussian noise is able to further improve the utility, except when the amount of noise is too big (e.g., $\sigma = 50$ and $100$) to ruin the model. We also indicate the privacy bound $\epsilon$ as accumulated by the moments accountant mechanism in (Abadi et al. [2016]). Interestingly, many $\epsilon$ values are too big to provide any meaningful privacy guarantee, but they are still able to improve the anomaly detection performance.

Besides AUPR scores, we further present AUROC scores in Table 5 for the same set of experiments. Autoencoders, as validated by many previous works (Gottschlich et al. [2017]), present great effectiveness in detecting outliers and novelties, especially when the outlier ratio in training dataset is slow (e.g., below $1\%$). Although not as obvious as AUPR scores, the improvements brought by differential privacy follow a similar trend, where the improvement is more significant with larger noise (smaller $\epsilon$) being applied to models trained with more outliers.

| noise scale $\sigma =$ | outlier percentage in training data $r_o$ | | | | | | | | | | | | $\epsilon$ |
|---|---|---|---|---|---|---|---|---|---|---|---|---|---|
| | 0.01% | | 0.1% | | 0.5% | | 1% | | 5% | | 10% | | |
| | OD | ND | OD | ND | OD | ND | OD | ND | OD | ND | OD | ND | |
| N/A | 100 | 99.84 | 99.92 | 99.77 | 92.12 | 98.81 | 92.12 | 99.81 | 84.33 | 88.18 | 72.16 | 68.14 | $\infty$ |
| 0 | 100 | 99.86 | 99.89 | 99.83 | 98.3 | 99.69 | 95.2 | 98.68 | 83.86 | 87.91 | 77.8 | 74.74 | $\infty$ |
| 0.001 | 100 | 99.70 | 100 | 99.86 | 98.37 | 99.64 | 94.33 | 98.81 | 84.31 | 89.34 | 86.51 | 85.58 | $1.0 \times 10^{10}$ |
| 0.005 | 100 | 99.87 | 99.82 | 99.78 | 98.69 | 99.7 | 95.67 | 98.87 | 91.01 | 94.3 | 79.55 | 77.04 | $3.9 \times 10^{8}$ |
| 0.01 | **100** | **99.89** | **100** | **99.97** | 94.92 | 99.23 | 97.08 | 99.33 | 90.79 | 93.34 | 85.41 | 84.07 | $9.8 \times 10^{7}$ |
| 0.05 | 100 | 99.85 | 99.89 | 99.79 | 97.97 | 99.55 | 96.94 | 99.34 | 88.84 | 92.54 | 75.18 | 72.09 | $2.8 \times 10^{6}$ |
| 0.1 | 100 | 99.88 | 100 | 99.85 | **98.44** | **99.66** | 93.11 | 98.21 | 92.23 | 94.21 | 85.56 | 83.98 | $6.8 \times 10^{4}$ |
| 0.5 | 100 | 99.84 | 99.95 | 99.85 | 98.59 | 99.64 | **99.95** | **99.85** | 93.69 | 95.8 | 85.61 | 83.86 | 22.23 |
| 1 | 100 | 99.81 | 100 | 99.78 | 98.28 | 99.67 | 95.8 | 99.0 | 94.92 | 96.87 | 81.87 | 80.12 | 3.09 |
| 5 | 100 | 99.49 | 99.87 | 99.49 | 98.51 | 99.52 | 96.5 | 98.78 | **96.78** | **98.04** | 95.25 | 95.41 | 0.44 |
| 10 | 97.62 | 97.61 | 90.24 | 97.77 | 91.88 | 98.2 | 97.5 | 99.12 | 96.6 | 98.2 | **97.07** | **97.46** | 0.25 |
| Below $\sigma$ value is too big such that the model does not converge well in training. | | | | | | | | | | | | | |
| 50 | 54.19 | 90.46 | 65.94 | 92.13 | 70.34 | 90.8 | 78.34 | 91.19 | 86.58 | 91.59 | 88.49 | 90.27 | 0.19 |
| 100 | 56.08 | 87.2 | 47.22 | 70.57 | 71.95 | 90.8 | 4.23 | 10.73 | 80.58 | 86.94 | 89.26 | 90.68 | 0.19 |

Table 4: AUPR scores for autoencoder outlier detection (OD) and novelty detection (ND).

| noise scale $\sigma =$ | outlier percentage in training data $r_o$ | | | | | | | | | | | | $\epsilon$ |
|---|---|---|---|---|---|---|---|---|---|---|---|---|---|
| | 0.01% | | 0.1% | | 0.5% | | 1% | | 5% | | 10% | | |
| | OD | ND | OD | ND | OD | ND | OD | ND | OD | ND | OD | ND | |
| N/A | 100 | 99.97 | 100 | 99.95 | 99.81 | 99.79 | 99.17 | 99.07 | 97.3 | 97.4 | 89.46 | 89.2 | $\infty$ |
| 0 | 100 | 99.97 | 100 | 99.97 | 99.96 | 99.93 | 99.7 | 99.77 | 97.36 | 97.52 | 92.39 | 92.35 | $\infty$ |
| 0.001 | 100 | 99.93 | 100 | 99.97 | 99.89 | 99.84 | 99.79 | 99.82 | 97.37 | 97.63 | 96.26 | 96.72 | $1.0 \times 10^{10}$ |
| 0.005 | 100 | 99.96 | 100 | 99.95 | 99.92 | 96.09 | 99.81 | 99.83 | 98.73 | 98.91 | 93.62 | 93.82 | $3.9 \times 10^{8}$ |
| 0.01 | 100 | 99.97 | 100 | 99.97 | 99.8 | 99.87 | 99.89 | 99.9 | 98.39 | 98.5 | 96.02 | 96.4 | $9.8 \times 10^{7}$ |
| 0.05 | 100 | 99.97 | 100 | 99.96 | 99.9 | 99.93 | 99.87 | 99.9 | 97.55 | 97.81 | 90.91 | 90.87 | $2.8 \times 10^{6}$ |
| 0.1 | 100 | 99.97 | 100 | 99.97 | 99.87 | 99.9 | 99.58 | 99.71 | 98.77 | 98.82 | 96.16 | 96.46 | $6.8 \times 10^{4}$ |
| 0.5 | 100 | 99.96 | 100 | 99.97 | 99.9 | 99.91 | 99.77 | 99.85 | 99.13 | 99.25 | 95.96 | 96.25 | 22.23 |
| 1 | 100 | 99.95 | 100 | 99.94 | 99.91 | 99.94 | 99.74 | 99.81 | 99.31 | 99.44 | 94.66 | 94.95 | 3.09 |
| 5 | 100 | 99.81 | 100 | 99.85 | 99.96 | 99.93 | 99.52 | 99.5 | 99.07 | 99.32 | 98.49 | 98.74 | 0.44 |
| 10 | 100 | 99 | 99.86 | 99.25 | 99.14 | 99.93 | 99.68 | 99.79 | 98.72 | 99.29 | 98.9 | 99.27 | 0.25 |
| Below $\sigma$ value is too big such that the model does not converge well in training. | | | | | | | | | | | | | |
| 50 | 99.99 | 96.11 | 94.8 | 96.41 | 92.96 | 96.13 | 94.15 | 96.29 | 7.03 | 5.95 | 93.83 | 95.61 | 0.19 |
| 100 | 99.73 | 94.7 | 82.64 | 87.29 | 93.51 | 99.22 | 31.77 | 30.91 | 94.47 | 96.08 | 94.62 | 96.09 | 0.19 |

Table 5: AUROC scores for autoencoder outlier detection (OD) and novelty detection (ND).

### A.2.2 Improvements over DeepLog

To compare the differentially private models with DeepLog, we utilize the same anomaly detection criteria, i.e., Top-$k$ based anomaly detection as what's presented in (Du et al. [2017]). Nevertheless, as a direct extension of the idea in measuring model loss for anomaly detection, we also tested the classification probability as one type of threshold for anomaly detection. In particular, if an actual system log entry is predicted with a probability lower than some threshold $T_p$, we treat this log entry as a detected anomaly. While the baseline DeepLog results are not as good as the Top-$k$ based detection, we show that similarly, differential privacy is able to significantly reduce the number of false negatives, without introducing too many new false positives.

**Probability-based detection.** As a preliminary result, we use probability-based anomaly detection to demonstrate the effectiveness of differential privacy noise in reducing FN . Table 6 shows FN and FP for DeepLog and DeepLog+DP with increasing noise levels, under different probability thresholds $T_p$. It is clear that differential privacy noise could effectively reduce FN , and the larger noise being added, the more false negatives are reduced. We also note that when $\sigma = 0.25$, the privacy bound $\epsilon = 90.5$. It is often thought that a privacy bound $\epsilon > 20$ is completely useless in terms of protecting privacy. Here we indicate that a small amount of noise may be enough to reduce FN . Although more false positives are incurred because of differential privacy noise, the drop in TPR is negligible, considering the large volume of normal data. For example, when $T_p = 2 \times 10^{-6}$ and $\sigma = 1$, the FN drop from 1261 to 183 indicates a TNR increase of 8% (91.7% $\rightarrow$ 98.8%), while the FP increase from 2291 to 3734 only shows a TPR decrease of 0.3% (99.59% $\rightarrow$ 99.32%).

| Probability threshold $T_p$ | DeepLog FN /FP | DeepLog+DP (FN /FP ) | | | |
|---|---|---|---|---|---|
| | | $\sigma = 0.25$ | $\sigma = 1$ | $\sigma = 1.5$ | $\sigma = 2$ |
| $10^{-5}$ | 573/3596 | 7/14268 | 0/6059 | 1/8213 | 1/9187 |
| $2 \times 10^{-6}$ | 1261/2291 | 208/4756 | 183/3734 | 1/5718 | 1/6317 |
| $10^{-6}$ | 1468/2068 | 410/3759 | 190/3552 | 2/4002 | 1/6093 |

Table 6: Probability-based anomaly detection results.

| | DeepLog | DeepLog+DP | | | | | | | |
|---|---|---|---|---|---|---|---|---|---|
| | | $\sigma$=0.25 | $\sigma$=0.5 | $\sigma$=0.75 | $\sigma$=1.0 | $\sigma$=1.25 | $\sigma$=1.5 | $\sigma$=1.75 | $\sigma$=2.0 |
| AUROC score | 0.9993 | 0.9997 | 0.9997 | 0.9997 | 0.9998 | 0.9993 | 0.9994 | 0.9989 | 0.9985 |
| privacy bound $\epsilon$ | 0 | 90.45 | 6.21 | 1.86 | 0.96 | 0.61 | 0.42 | 0.31 | 0.25 |

Table 7: AUROC score comparison and privacy bound $\epsilon$.

**AUROC score**    To evaluate the overall performance of DeepLog+DP compared with DeepLog under different thresholds, we further compute the AUROC score of DeepLog and DeepLog+DP with different noise scale $\sigma$. As shown in Table 7, DeepLog already achieves excellent AUROC score, considering the large amount of normal data and the significantly fewer anomalies. However, an adequate amount of differential privacy noise is still able to improve the performance.

**Privacy bound** $\epsilon$    Table 7 also indicates the privacy bound $\epsilon$. Note that $\epsilon < 10$ is often considered as usable and $\epsilon < 1$ is a tight bound that well protects privacy. Considering all the cases, $\sigma = 1$ gives the best anomaly detection utility as well as a tight privacy bound to protect training data privacy.

### A.2.3 Backdoor attack detection

In this section, we evaluate more parameters for the experiment set up described in Section 4.3 BACKDOOR ATTACK DETECTION, and measure *benign accuracy*, *success rate*, AUPR score and AUROC score as explained in Section 4.3 for each experiment setting. Similar as the observations in Section 4.3, a differentially private trained machine learning model is naturally more robust to backdoor attacks. The evidence is that the benign accuracy (Table 8) is affected little by differential privacy except when the noise scale is too big to ruin the model parameters, compared with the significant downgrade (e.g., 98.1% to 0.3%) in backdoor success rate (Table 9). Also, as shown in Table 10 and Table 11, measuring model loss to detect poisoning examples could be useful when the

poisoning ratio is low. Nevertheless, applying differential privacy is able to significantly improve the detection performance for a poisoning ratio as high as $45\%$.

| noise scale | poisoning ratio in training data $r_p$ | | | | | | | | | $\epsilon$ |
|---|---|---|---|---|---|---|---|---|---|---|
| $\sigma$ | 0.005 | 0.01 | 0.05 | 0.1 | 0.2 | 0.3 | 0.4 | 0.45 | | |
| N/A | 98.93 | 99.03 | 98.95 | 99.11 | 98.94 | 99.06 | 99.05 | 98.97 | | $\infty$ |
| 0 | 97.66 | 97.21 | 97.84 | 97.46 | 96.97 | 96.32 | 93.61 | 92.4 | | $\infty$ |
| 0.001 | 97.5 | 97.52 | 97.72 | 97.29 | 97.47 | 96.69 | 94.16 | 90.41 | | $9.9 \times 10^9$ |
| 0.005 | 97.57 | 97.55 | 97.46 | 97.75 | 97.36 | 96.5 | 93.96 | 91.25 | | $3.9 \times 10^8$ |
| 0.01 | 97.51 | 97.61 | 97.4 | 97.55 | 97.27 | 97.07 | 94.77 | 92.82 | | $9.8 \times 10^7$ |
| 0.05 | 97.42 | 97.87 | 97.8 | 97.72 | 97.69 | 96.37 | 94.19 | 93.28 | | 2830766.11 |
| 0.1 | 97.55 | 97.84 | 97.7 | 97.34 | 97.29 | 96.91 | 94.11 | 91.36 | | 67915.88 |
| 0.5 | 97.56 | 97.29 | 97.28 | 97.37 | 97.13 | 96.55 | 95.19 | 86.37 | | 22.23 |
| 1 | 96.94 | 96.95 | 96.96 | 96.53 | 96.27 | 95.78 | 91.65 | 83.16 | | 3.09 |
| 2 | 93.76 | 93.39 | 92.16 | 93.22 | 92.77 | 92.4 | 87.61 | 81.24 | | 1.18 |
| Below noise level $\sigma$ could be too high. | | | | | | | | | | |
| 3 | 89.85 | 89.5 | 91.12 | 90.92 | 89.35 | 89.81 | 85.42 | 76.51 | | 0.75 |
| 5 | 80.51 | 79.57 | 80.49 | 80.28 | 79.19 | 77.95 | 57.5 | 61.64 | | 0.44 |
| 10 | 17.32 | 19.82 | 20.31 | 12.07 | 11.34 | 11.43 | 11.15 | 11.11 | | 0.25 |

Table 8: Benign accuracy of models trained on datasets with different poisoning ratio $r_p$. The more noise being added, the more utility is affected.

| noise scale | poisoning ratio in training data $r_p$ | | | | | | | | | $\epsilon$ |
|---|---|---|---|---|---|---|---|---|---|---|
| $\sigma$ | 0.005 | 0.01 | 0.05 | 0.1 | 0.2 | 0.3 | 0.4 | 0.45 | | |
| N/A | 47.85 | 90.96 | 97.12 | 98.1 | 98.46 | 98.91 | 98.92 | 98.79 | | $\infty$ |
| 0 | 0.23 | 0.29 | 0.35 | 0.3 | 0.47 | 0.74 | 68.1 | 72.02 | | $\infty$ |
| 0.001 | 0.21 | 0.22 | 0.25 | 0.37 | 0.42 | 0.85 | 35.83 | 19.01 | | $9.9 \times 10^9$ |
| 0.005 | 0.17 | 0.2 | 0.28 | 0.3 | 0.35 | 35.54 | 50.28 | 83.94 | | $3.9 \times 10^8$ |
| 0.01 | 0.25 | 0.24 | 0.34 | 0.31 | 0.42 | 0.56 | 93.64 | 15.25 | | $9.8 \times 10^7$ |
| 0.05 | 0.24 | 0.25 | 0.37 | 0.3 | 0.37 | 18.55 | 82.15 | 4.53 | | 2830766.11 |
| 0.1 | 0.25 | 0.18 | 0.28 | 0.39 | 0.47 | 0.71 | 82.23 | 59.57 | | 67915.88 |
| 0.5 | 0.26 | 0.23 | 0.29 | 0.35 | 0.37 | 0.81 | 1.63 | 74.15 | | 22.23 |
| 1 | 0.28 | 0.3 | 0.45 | 0.5 | 0.63 | 1.09 | 44.12 | 67.44 | | 3.09 |
| 2 | 0.74 | 0.68 | 1.07 | 1.1 | 1.4 | 2.67 | 5.95 | 71.48 | | 1.18 |
| Below noise level $\sigma$ could be too high. | | | | | | | | | | |
| 3 | 0.96 | 1.22 | 1.1 | 1.59 | 2.69 | 2.66 | 6.02 | 20.38 | | 0.75 |
| 5 | 2.01 | 1.6 | 2.44 | 3.47 | 2.68 | 8.15 | 13.78 | 19.91 | | 0.44 |
| 10 | 9.93 | 10.3 | 10.03 | 9.08 | 9.68 | 9.33 | 10.14 | 9.77 | | 0.25 |

Table 9: Backdoor attack success rate of models trained on datasets with different poisoning ratio $r_p$. The success rate is significantly reduced for models trained with differential privacy.

| noise scale | poisoning ratio in training data $r_p$ | | | | | | | | $\epsilon$ |
|---|---|---|---|---|---|---|---|---|---|
| $\sigma$ | 0.005 | 0.01 | 0.05 | 0.1 | 0.2 | 0.3 | 0.4 | 0.45 | |
| N/A | 73.01 | 27.02 | 14.85 | 17.63 | 24.9 | 36.42 | 42.16 | 45.85 | $\infty$ |
| 0 | 91.22 | 92.11 | 95.33 | 95.46 | 95.9 | 96.55 | 62.57 | 60.33 | $\infty$ |
| 0.001 | 91.52 | 93.36 | 94.61 | 95.98 | 96.95 | 96.05 | 78.53 | 86.9 | $9.9 \times 10^9$ |
| 0.005 | 92.64 | 94.04 | 94.76 | 95.45 | 96.98 | 79.1 | 72.73 | 52.07 | $3.9 \times 10^8$ |
| 0.01 | 92.24 | 94.03 | 93.4 | 95.76 | 96.69 | 96.22 | 47.08 | 90.77 | $9.8 \times 10^7$ |
| 0.05 | 90.76 | 95.11 | 95.09 | 95.54 | 96.35 | 87.28 | 49.82 | 93.72 | 2830766.11 |
| 0.1 | 92.16 | 94.85 | 95.33 | 95.28 | 96.4 | 96.67 | 51 | 65.9 | 67915.9 |
| 0.5 | 92.76 | 93.4 | 94.5 | 94.93 | 95.74 | 95.88 | 95.96 | 54.99 | 22.23 |
| 1 | 88.67 | 90.31 | 94.77 | 94.46 | 96.03 | 94.67 | 72.99 | 56.35 | 3.09 |
| 2 | 65.01 | 78.51 | 80.76 | 87.54 | 88.76 | 87.63 | 86.75 | 51.41 | 1.18 |
| Below noise level $\sigma$ could be too high. | | | | | | | | | |
| 3 | 29.31 | 51.37 | 78.25 | 81 | 79.83 | 81.56 | 84.46 | 71.62 | 0.75 |
| 5 | 6.4 | 17.39 | 58.66 | 58.88 | 61.8 | 69.19 | 64.27 | 61.58 | 0.44 |
| 10 | 0.8 | 2 | 10.12 | 10.88 | 20.32 | 30.89 | 40.92 | 45.17 | 0.25 |

Table 10: AUPR scores for backdoor attack detection. Applying differential privacy significantly improves the results.

| noise scale | poisoning ratio in training data $r_p$ | | | | | | | | $\epsilon$ |
|---|---|---|---|---|---|---|---|---|---|
| $\sigma$ | 0.005 | 0.01 | 0.05 | 0.1 | 0.2 | 0.3 | 0.4 | 0.45 | |
| N/A | 99.26 | 95.23 | 78.88 | 67.72 | 59.47 | 60.9 | 55.33 | 52.73 | $\infty$ |
| 0 | 99.92 | 99.88 | 99.79 | 99.72 | 99.51 | 99.31 | 70.24 | 62.88 | $\infty$ |
| 0.001 | 99.91 | 99.88 | 99.79 | 99.72 | 99.62 | 99.27 | 81.27 | 88.1 | $9.9 \times 10^9$ |
| 0.005 | 99.9 | 99.93 | 99.79 | 99.75 | 99.63 | 84.51 | 75.12 | 60.21 | $3.9 \times 10^8$ |
| 0.01 | 99.92 | 99.92 | 99.74 | 99.74 | 99.61 | 99.3 | 61.83 | 93.78 | $9.8 \times 10^7$ |
| 0.05 | 99.9 | 99.94 | 99.83 | 99.73 | 99.59 | 90.67 | 58.03 | 97.12 | 2830766.11 |
| 0.1 | 99.93 | 99.93 | 99.82 | 99.69 | 99.55 | 99.37 | 59.71 | 65.55 | 67915.88 |
| 0.5 | 99.95 | 99.92 | 99.76 | 99.68 | 99.5 | 99.23 | 98.68 | 58.44 | 22.23 |
| 1 | 99.86 | 99.84 | 99.76 | 99.61 | 99.43 | 98.87 | 75.7 | 63.94 | 3.09 |
| 2 | 99.64 | 99.64 | 98.97 | 98.86 | 98.21 | 97.13 | 94.82 | 57.81 | 1.18 |
| Below noise level $\sigma$ could be too high. | | | | | | | | | |
| 3 | 98.68 | 98.92 | 98.68 | 98.17 | 96.73 | 95.92 | 93.86 | 81.22 | 0.75 |
| 5 | 95.74 | 96.13 | 96.38 | 94.42 | 92.22 | 89.52 | 77.77 | 73.11 | 0.44 |
| 10 | 56.77 | 60.49 | 61.2 | 53.11 | 51.05 | 51.33 | 51.19 | 50.35 | 0.25 |

Table 11: AUROC scores for backdoor attack detection. It shows that measuring model loss for poisoning samples detection could be effective when the poisoning ratio is low. Differential privacy improves the performance in all cases, except when the noise scale is too big to ruin the model parameters.

