# OpenReview forum: "Robust anomaly detection and backdoor attack detection via differential privacy"
_ICLR.cc/2020/Conference — Accept (Poster)_

### Official Review · AnonReviewer3 · 2019-10-24
**Official Blind Review #3**

**Rating:** 3

**Review:**

Interesting topic but lacks of novelty
#Summary:
The paper proposes that by applying differential privacy, the performance on outlier and novelty detection can be improved. It first presents a theoretical analysis, which establishes a lower bound on the prediction performance difference between normal and outlier data. By adding noise into the training process, the outliers in the dataset will be hidden by the noise, which will result in a model that utilizes the normal data. In this way, when deploying the model, the model will find the outlier by observing low confidence.

#Strength
It is good to see that the paper builds a connection between the privacy parameter and the noise level and the experiments make the theory valid.

#Weakness
I’m not an expert in differential privacy. But as far as I’m concerned, a typical downside is that the false positive rate will increase and there is no theoretical guarantee that the increase of false-positive rate will be negligible compared with the increase of true positive rate.
Its effectiveness in detecting backdoor attacks seems elusive. As we know, the backdoor attacks exist when users want to outsource the task of training the network to a third-party, which may potentially be an attack. Therefore, the training process is out-of-control to the detector. However, the paper proposes to use differential privacy to the model training process, which is not in the settings of a backdoor attack.


**Experience Assessment:**

I do not know much about this area.

**Review Assessment: Checking Correctness Of Derivations And Theory:**

I assessed the sensibility of the derivations and theory.

**Review Assessment: Checking Correctness Of Experiments:**

I assessed the sensibility of the experiments.

**Review Assessment: Thoroughness In Paper Reading:**

I read the paper at least twice and used my best judgement in assessing the paper.

---

> ### Author Response · Authors · 2019-11-12
> **Response to Reviewer #3**
>
> Response to R3:
>
> Thank you for the valuable comments!
>
> Q: I’m not an expert in differential privacy. But as far as I’m concerned, a typical downside is that the false positive rate will increase and there is no theoretical guarantee that the increase of false-positive rate will be negligible compared with the increase of true positive rate.
> Its effectiveness in detecting backdoor attacks seems elusive. As we know, the backdoor attacks exist when users want to outsource the task of training the network to a third-party, which may potentially be an attack. Therefore, the training process is out-of-control to the detector. However, the paper proposes to use differential privacy to the model training process, which is not in the settings of a backdoor attack.
>
> A: First of all, we want to note that although false positives may slightly increase with DP noise added, false negatives reduces significantly, and the overall F-measure also increases a lot. As shown in Figure 2(b), the best F-measure of DeepLog with DP (96.29%) is well above the best F-measure of DeepLog without DP (93.66%). With a reasonable noise scale (not too large), DeepLog with DP almost always outperforms the state-of-the-art real-world system log anomaly detection model in computer security domain in terms of various metrics that take into account both false negatives and false positives. Also, as indicated in the first paragraph of page 7, reducing false negatives could be more important because false positives could be further checked by system admin, while false negatives may never be discovered until a disastrous event occurs.
>
> Secondly, the backdoor attack scenario mentioned by the reviewer is indeed a common one. Another common scenario for backdoor attacks is crowdsourcing, where the model trainer gathers training data from untrusted individuals. In this case the model trainer does not have control over the data quality but does have control over the model training process. Our proposal of adding DP noise is useful for detecting backdoor attacks and training more robust models in such scenario. We have clarified the use case of our method in the first paragraph of Section 4.3.
>
> We will appreciate it if the reviewer has further comments!

---

### Official Review · AnonReviewer1 · 2019-10-27
**Official Blind Review #1**

**Rating:** 6

**Review:**

This paper proposes the idea of using differential privacy (DP) to improve the performance of outlier and novelty detection. Differential privacy was proposed as a privacy metric which limits the contribution of a single data point in the training set to the output. This property naturally controls how poisoned data would affect the output of the learned model. Under the assumption that a well-trained model would incur a higher loss on the outliers, the paper gives a theoretic bound on how this loss will decrease if there are poisoned samples in the training set.

The paper also performs several experiments on synthetic and real-world datasets. The paper shows that add differential privacy during training can improve the performance of autoencoder-based outlier detection on MNIST data.  For real-world data, the paper improves the performance of anomaly detection on the HDFS dataset over the state-of-the-art algorithm. The paper also shows empirically how DP can help improve backdoor attack detection.

The paper is overall nicely written with some nice results. The paper could be improved if the following confusions can be resolved.

1. Novelty detection is generally referred to as detecting samples in the test set that are not in the distribution of the training set. In the theory part, the analysis is mostly based on data poisoning, which is not typical in the novelty detection setting. I hope this can be clarified.
2. In the experiment part, the paper uses Figure 1 to show how UAERM is satisfied. I find this a bit confusing. In definition 4, the h^* is referred to as the global minimizer while in the experiment, the empirical minimizer is used.
3. Theorem 2 presents some theoretical bound to show the power of DP on improving outlier detection, however, in the parameter setting used in the experiment, Theorem 2 does not provide meaningful bounds. There is a bit disconnection between the two parts.

Based on the above comments, I think the paper can be accepted if there is room for it. But I won't push it for acceptance.

**Experience Assessment:**

I have read many papers in this area.

**Review Assessment: Checking Correctness Of Derivations And Theory:**

I assessed the sensibility of the derivations and theory.

**Review Assessment: Checking Correctness Of Experiments:**

I assessed the sensibility of the experiments.

**Review Assessment: Thoroughness In Paper Reading:**

I read the paper at least twice and used my best judgement in assessing the paper.

---

> ### Author Response · Authors · 2019-11-12
> **Response to Reviewer #1**
>
> Response to R1:
>
> Thank you very much for your kind review and helpful comments!
>
> Q: Novelty detection is generally referred to as detecting samples in the test set that are not in the distribution of the training set. In the theory part, the analysis is mostly based on data poisoning, which is not typical in the novelty detection setting. I hope this can be clarified.
>
> A: We apologize for not making the implication of our theory as clear as we intended. Our theory can be applied to novelty detection because our definition of outliers only says that their learning loss deviates from the loss of normal distribution data. Such outliers could be in train set or test set. We have further clarified this at the last paragraph of Section 3 in our revised paper.
>
> Q: In the experiment part, the paper uses Figure 1 to show how UAERM is satisfied. I find this a bit confusing. In definition 4, the h^* is referred to as the global minimizer while in the experiment, the empirical minimizer is used.
>
> A:  We apologize for the confusion. Indeed, our experiment aims to perform a sanity check of the UAERM assumption, as the rigorous verification of the assumption requires evaluating the expected loss over the data distribution, which is simply impossible as the true data distribution is unknown. Therefore, in the experiments, we replace the expectation by the empirical average of random samples. We have explained more about this set of experiments in Section 4.1 "Validation of UAERM for Noisy SGD".
>
> Q: Theorem 2 presents some theoretical bound to show the power of DP on improving outlier detection, however, in the parameter setting used in the experiment, Theorem 2 does not provide meaningful bounds. There is a bit disconnection between the two parts.
>
> A:  We agree with the reviewer that our bound is not tight; however, our theory is still *useful* as it can be used to explain various trends in our experiments.  For example, as shown in Table 1, the more outliers in training dataset, the higher noise scale it requires to achieve the best anomaly detection performance. This scenario can be explained by Theorem 2. As the second paragraph on page 4 explains, our theory shows that the privacy parameters cannot be too large or too small to ensure optimal anomaly detection performance, which coincides with the experimental results in Table 1. We have revised the paper to clarify the correlation between the implications of our theory and experimental findings, which could be found in the last paragraph of Section 4.1.

---

### Official Review · AnonReviewer2 · 2019-11-04
**Official Blind Review #2**

**Rating:** 6

**Review:**

This paper leverages differential privacy’s stability properties to investigate its use for improved anomaly and backdoor attack detection. Under an assumption (called “uniformly asymptotic empirical risk minimization”), the authors show that difference between the expected loss of a differentially private learning algorithm on an outlier (where the expectation is taken over the randomness of the learning algorithm) and the expected loss of the same algorithm on data from the underlying distribution (expectation taken over data & randomness of the algorithm) is lower bounded by a (possibly/hopefully) non-negative quantity with high probability. The authors then conduct a set of experiments to show that differential privacy improves the performance of outliers, novel examples, and backdoor attack detection.

Overall, the paper is very well written and easy to read. The paper also tackles an important and timely problem that is relevant to the ICLR community. While there has been some recent work on connecting differential privacy to robustness & attacks, this paper investigates the use of differential private model training as a means to improve novelty detection at inference time.

A few points that need attention from the authors:

1. The theory developed is insightful in general but has very little (to no) practical value. For starters, it assumes that differentially private model training is uniformly asymptotic to empirical risk minimization. This is not necessarily true for highly non-convex models trained with SGD. Further, it cannot be verify via experimentations (despite the authors’ attempt to sanity check it using Figure 1). More importantly, the theory developed in Section 3 is not used in any meaningful way in the experiments section — the anomaly detection schemes are agnostic to it.
2. The authors make no attempt to co-optimize the performance of the model with its ability to be used for better anomaly detection. For instance, the authors choose an l2-clipping-norm C of 1 and do not consider trading off C with the noise variance.

When the training set contains anomalies, this work can be viewed as “what is the impact of differential privacy” on a training sets with a majority group (training examples from a given distribution) and a minority group (training examples from a different distribution). Under this view, this paper essentially says that “differential privacy leads to disparate impact on model accuracy/loss”. This has been recently investigated in the following NeurIPS19 paper: https://arxiv.org/abs/1905.12101. Thus the contributions of the paper are not substantial.


**Experience Assessment:**

I have published one or two papers in this area.

**Review Assessment: Checking Correctness Of Derivations And Theory:**

I carefully checked the derivations and theory.

**Review Assessment: Checking Correctness Of Experiments:**

I carefully checked the experiments.

**Review Assessment: Thoroughness In Paper Reading:**

I read the paper thoroughly.

---

> ### Author Response · Authors · 2019-11-12
> **Response to Reviewer #2 (part 1)**
>
> Thank you for your valuable comments and for pointing out the related work, which have greatly helped to improve the paper.
>
> Q: The theory developed is insightful in general but has very little (to no) practical value. For starters, it assumes that differentially private model training is uniformly asymptotic to empirical risk minimization. This is not necessarily true for highly non-convex models trained with SGD. Further, it cannot be verify via experimentations (despite the authors’ attempt to sanity check it using Figure 1). More importantly, the theory developed in Section 3 is not used in any meaningful way in the experiments section — the anomaly detection schemes are agnostic to it.
>
> A: We agree with the reviewer that the rigorous verification of the UAERM assumption for deep neural network cannot be done empirically, as it involves evaluating the expectation of loss taken over the randomness of differentially private algorithms as well as randomness of data distribution. The role of Figure 4 is indeed a sanity check of the assumption by replacing the expectation with the average of random samples. The results in Figure 4 demonstrate the plausibility of the assumption. We have added the explanation as well as the detailed experiment setup in Section 4.1 "Validation of UAERM for Noisy SGD".
>
> Moreover, we would like to emphasize that our theory is *useful* as it can be used to explain various findings in the experiments. For example, as shown in Table 1, the more outliers in the training dataset, the higher noise scale it requires to achieve the best anomaly detection performance. This scenario can be explained by Theorem 2. As the second paragraph on page 4 explains, our theory shows that the privacy parameters cannot be too large or too small to ensure optimal anomaly detection performance, which coincides with the experimental results in Table 1. We have revised the paper to clarify the correlation between the implications of our theory and experimental findings, which could be found in the last paragraph of Section 4.1.
>
> Finally, our theory is a generic result and it can be used to explain the performance to detect general out-of-distribution samples. Anomaly detection is one of the applications that our theory can apply to. We have also added a detailed explanation in the last paragraph of Section 3.
>
> We will appreciate it if the reviewer has further suggestions.
>
> Q: The authors make no attempt to co-optimize the performance of the model with its ability to be used for better anomaly detection. For instance, the authors choose an l2-clipping-norm C of 1 and do not consider trading off C with the noise variance.
>
> A:  Thank you for pointing out the direction of co-optimizing the performance of the model with its ability to be used for better anomaly detection. We apologize if our paper didn’t explain well. In fact, for all 3 models evaluated in the paper,  including the autoencoders and DeepLog, their sole purpose is anomaly detection. Only for backdoor attack detection did we use the same CNN model for both image classification and backdoor detection. We have added the experiments of varying C with fixed noise variance for backdoor attack detection, which could be found in Table 3. From Table 2&3, we can observe that the best model for anomaly detection could have a similar set of parameters with the best model for image classification. However, in general, classification accuracy and robustness are two conflicting desiderata; model trainers can tune the privacy parameter in order to meet the task-specific requirements for accuracy and robustness. We have added the discussion of the results in the last paragraph of Section 4.3.

---

> > ### Author Response · Authors · 2019-11-15
> > **Response to Reviewer #2 (part 2)**
> >
> > Q: When the training set contains anomalies, this work can be viewed as “what is the impact of differential privacy” on a training sets with a majority group (training examples from a given distribution) and a minority group (training examples from a different distribution). Under this view, this paper essentially says that “differential privacy leads to disparate impact on model accuracy/loss”. This has been recently investigated in the following NeurIPS19 paper: https://arxiv.org/abs/1905.12101. Thus the contributions of the paper are not substantial.
> >
> > A: Thank you for pointing out the reference! It is indeed very relevant and discovered the same intrinsic phenomenon as us. In our revised paper, we have cited this paper and discussed more in the last paragraph of Section 5 Related work. In short, while the related paper explains the phenomenon in a more generic way, our work includes a theoretical justification, which, for the first time, precisely characterizes the dependence of the performance gap between the majority and the minority group on the privacy parameters. Our theory is further backed up by extensive experiments in anomaly detection, novelty detection and backdoor attack detection; notably, our proposed method has greatly improved the state-of-the-art system log anomaly detection performance, which is itself a significant contribution to the computer security area. By contrast, the reference mainly considered the implication of differential privacy to the fairness of machine learning models.

---

### Decision · Program_Chairs · 2019-12-19

**Decision:**

Accept (Poster)

**Comment:**

Thanks for the submission. This paper leverages the stability of differential privacy for the problems of anomaly and backdoor attack detection. The reviewers agree that this application of differential privacy is novel. The theory of the paper appears to be a bit weak (with very strong assumptions on the private learner), although it reflects the basic underlying idea of the detection technique. The paper also provides some empirical evaluation of the technique.